# Cafe-Talk: Generating 3D Talking Face Animation with Multimodal Coarse- and Fine-grained Control

**Hejia Chen**[1,♣,♠]    **Haoxian Zhang**[2,♣]    **Shoulong Zhang**[3,♣]    **Xiaoqiang Liu**[2]
**Sisi Zhuang**[1]    **Yuan Zhang**[2]    **Pengfei Wan**[2]    **Di Zhang**[2]    **Shuai Li**[1,3,♡]

[1]State Key Laboratory of Virtual Reality Systems and Technology, Beihang University
[2]Kuaishou Technology    [3] Zhongguancun Laboratory

♣Equal contribution    ♠Intern at Kuaishou Technology    ♡Corresponding author

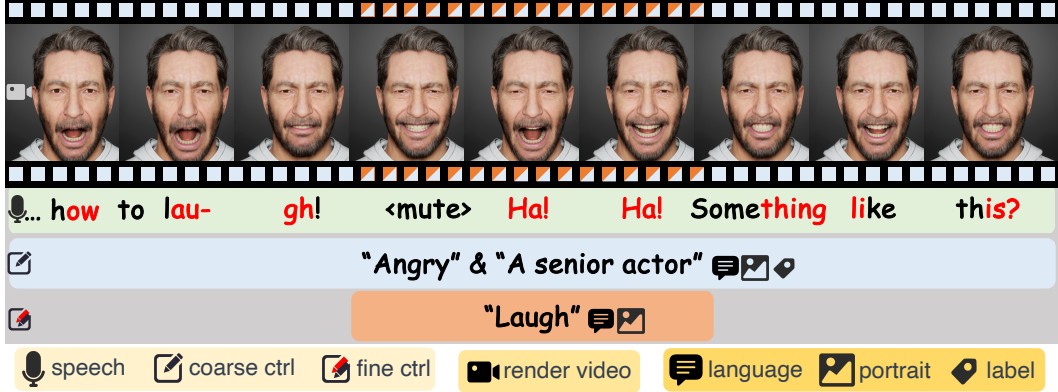

Figure 1: Adding multimodal coarse- and fine-grained control enables more flexible animations: **Scenario**: A senior actor is arguing with the director about how to smile. **Action**: The actor responds with anger and concludes with a sudden sarcastic laugh.

## Abstract

Speech-driven 3D talking face method should offer both accurate lip synchronization and controllable expressions. Previous methods solely adopt discrete emotion labels to globally control expressions throughout sequences while limiting flexible fine-grained facial control within the spatiotemporal domain. We propose a diffusion-transformer-based 3D talking face generation model, *Cafe-Talk*, which simultaneously incorporates coarse- and fine-grained multimodal control conditions. Nevertheless, the entanglement of multiple conditions challenges achieving satisfying performance. To disentangle speech audio and fine-grained conditions, we employ a two-stage training pipeline. Specifically, Cafe-Talk is initially trained using only speech audio and coarse-grained conditions. Then, a proposed fine-grained control adapter gradually adds fine-grained instructions represented by action units (AUs), preventing unfavorable speech-lip synchronization. To disentangle coarse- and fine-grained conditions, we design a swap-label training mechanism, which enables the dominance of the fine-grained conditions. We also devise a mask-based CFG technique to regulate the occurrence and intensity of fine-grained control. In addition, a text-based detector is introduced with text-AU alignment to enable natural language user input and further support multimodal control. Extensive experimental results prove that Cafe-Talk achieves state-of-the-art lip synchronization and expressiveness performance and receives wide acceptance in fine-grained control in user studies. Project page: https://harryxd2018.github.io/cafe-talk/

# 1 INTRODUCTION

Generating realistic speech-driven 3D facial animation holds significant application value in the traditional film industries and advanced AI applications. It is a challenging task that necessitates synchronized lip motions and the conveyance of vivid, compelling facial expressions. Moreover, an animator-friendly and agent-driven application, as illustrated in Fig. 1, requires fine-grained and multimodal control capabilities. Although existing methods have achieved impressive progress in facial expression control by using emotion labels (Daněček et al., 2023; Haque & Yumak, 2023) or multimodal emotional references (Xu et al., 2023; Peng et al., 2023b) for entire sequences, modeling and flexibly controlling fine-grained facial expressions during the speech is still unexplored.

Our motivation is to present a 3D talking face method with effective, precise, and flexible controllability. To achieve this, the controlling condition is spatiotemporally modeled as coarse- and fine-grained conditions, where the coarse-grained condition establishes a foundation for the produced facial movements and the fine-grained condition enriches the details. Spatially, the coarse-grained conditions abstractly depict overall facial movements through talking style and emotion. Conversely, the fine-grained condition emphasizes specific and localized muscle movements, such as blinking and raising the eyebrows. Temporally, the coarse-grained condition remains constant during speaking, while the fine-grained condition describes the instant facial expressions. Existing methods (Daněček et al., 2023; Peng et al., 2023b) focus on modeling the talking style and emotion temporal-consistently, neglecting fine-grained facial movements in spatiotemporal domains, which results in an inability to control intricate local motions and flexible expression changes.

To tackle the aforementioned challenge, in this paper, we present Cafe-Talk as the first 3D talking face generation method with **coa**rse- and **fine**-grained multimodal control. In order to achieve detailed and flexible control within the current diffusion-based framework, an intuitive approach is to directly implement additional fine-grained conditions and jointly train the network alongside speech audio and coarse-grained conditions.

However, the entanglement of the three conditions results in unsynchronized lip movements and an inability to fine-grained control based on our preliminary experimental findings. Specifically, (1) the entanglement of the audio and fine-grained condition provides a shortcut to producing lip movements and weakening speech audio guidance, as the fine-grained conditions can also describe local muscle motions around the mouth. To this end, Cafe-Talk is designed as a coarse-to-fine structure with a two-stage pipeline. In the first stage, a diffusion-based transformer (Ng et al., 2024; Zhao et al., 2024) generates the facial movements with encoded speech audio and incorporates the multimodal coarse-grained condition following Peebles & Xie (2023) to achieve efficient global control. In the second stage, the fine-grained conditions represent a sequence of action units (AUs), which flexibly guide the corresponding muscles to stay excited. Although existing methods (Ma et al., 2023; Wang et al., 2024; Sun et al., 2024) utilize AUs to control facial motions, they merely refine the coarse-grained control instead of disentangle the detail controllability. With the frozen base model, we design a fine-grained control adapter to gradually add the fine-grained condition into the pipeline without jeopardizing lip movements benefiting from a zero convolution layer (Zhang et al., 2023a). Moreover, (2) to achieve effective, precise, and flexible fine-grained control, we first build a swap-label training mechanism upon the two-stage pipeline to introduce conflict coarse- and fine-grained conditions, enabling predominant fine-grained conditional control. We then design a mask-based classifier-free guidance (CFG) technique for inference, which ensures the occurrence of fine-grained conditions and allows for effective and accurate intensity control. We finally introduce a text-based detector for language-based emotion and expression description by aligning it with AUs on a projected CLIP space (Radford et al., 2021), extending flexible multimodal fine-grained control.

We thoroughly validate the effectiveness of our proposed controllable talking face generation method through extensive experiments. To the best of our knowledge, we are the first to address the fine-grained 3D talking face control task. Thus, we compare the lip synchronization and coarse-grained expression control capabilities with existing methods and achieve state-of-the-art performances. Additionally, we conduct ablation experiments to demonstrate the importance of each proposed module in achieving fine-grained control. Finally, we execute a user study and received wide acceptance of the effectiveness of our control. The contributions of this paper can be summarized as follows:

1. We propose Cafe-Talk, the first 3D talking face generation model enabling coarse- and fine-grained multimodal conditional control in an effective, precise, and flexible manner.

2. We design a two-stage training pipeline that explicitly disentangles speech audio and fine-grained conditions. Additionally, we devise a fine-grained control adapter that gradually introduces fine-grained conditions without jeopardizing lip synchronization.

3. We introduce a swap-label training mechanism and mask-based CFG technique in the inference process to disentangle coarse- and fine-grained conditions, which achieves dominant and precise localized facial control.

4. We devise a text-based AU detector to efficiently extract AU fine-grained conditions from natural language descriptions based on user intent, which enables our model to support multimodal control.

## 2 RELATED WORK

Talking face animation has attracted considerable attention due to its significant application potential and market demand. This technology can be broadly divided into two categories based on the output format: video-based and 3D-based methods. Video-based methods (Zhou et al., 2020; Wang et al., 2020; Liu et al., 2023; Guo et al., 2024) focus on generating realistic portrait videos, particularly benefiting from advanced neural rendering techniques (Guo et al., 2021; Shen et al., 2022; Ye et al., 2023; Peng et al., 2024). In contrast, this paper focuses on 3D-based methods, which produce 3D meshes or 3DMM coefficients that are compatible with academic 3D face reconstruction methods (Bao et al., 2021; Chai et al., 2022; Bai et al., 2023) and industrial game engines, offering broader practical applications. Within the domain of 3D-based methods, previous works (Cudeiro et al., 2019; Fan et al., 2022; Xing et al., 2023; Yi et al., 2023; Bao et al., 2023; Peng et al., 2023a) leveraging discriminative models have successfully achieved precise lip movement regression from speech audio. These methods typically use a pretrained speech audio encoder (e.g., from automatic speech recognition models (Hannun et al., 2014; Baevski et al., 2020)) and a decoder to translate audio features into lip movements. While these methods effectively synchronize lip movements, the facial expressions generated are often lacking in diversity and emotional expressiveness. With the advancement of generative models, recent methods (Aneja et al., 2024; Ng et al., 2024) have constructed diffusion-based pipelines to achieve more diverse and accurate facial movements. To improve expressiveness in talking face animation, some methods extract emotion cues from the input speech. For example, EmoTalk (Peng et al., 2023b) disentangles content and emotion from the speech audio using cross-reconstruction techniques. However, emotion cues derived from speech are often person-specific, which limits the generalization capability of such methods. An alternative approach involves incorporating explicit emotion labels into the talking face animation pipeline, as seen in (Karras et al., 2017; Haque & Yumak, 2023; Daněček et al., 2023). These methods, trained on datasets with emotion annotations, enable better control over expressiveness. Unlike previous methods that use one-hot encoding for emotion labels, Xu et al. (2023) proposed encoding emotion annotations using CLIP, leveraging its semantic space to model unseen emotions, thus enabling more flexible control.

Fine-grained control in body motion generation has been widely studied, evolving from generating motions based on single actions (Petrovich et al., 2021) to more sophisticated control using natural language (Tevet et al., 2022; 2023). Some works (Wang et al., 2023; Lu et al., 2024) have focused on enhancing the spatial precision of motion described by text, while others (Zhang et al., 2023b; Li et al., 2023; Zhang et al., 2024; Petrovich et al., 2024) model the temporal aspects of motion to generate coherent sequences for motion control. Inspired by these advancements, recent methods have extended emotion control in talking face animation from label-based approaches to fine-grained control. Methods (Ma et al., 2023; Gan et al., 2023; Tan et al., 2024; Wang et al., 2024; Zhao et al., 2024; Sun et al., 2024) have explored generating talking face animations using emotion or expression descriptions. In these works, AUs (Friesen & Ekman, 1978) have been frequently used to create precise descriptions of facial movements — TalkCLIP (Ma et al., 2023) obtained rule-based expression descriptions by summarizing frequently activated AUs from ground truth videos, alongside the corresponding emotion category and intensity from datasets (Wang et al., 2020). Style$^2$Talker (Tan et al., 2024) and InstructAvatar (Wang et al., 2024) further improved these descriptions with the help of large language models (LLMs). However, due to the entanglement between emotion labels and AUs, these methods only refine emotion labels using AUs rather than providing temporally

fine-grained control. To enable multimodal control, TalkCLIP aligned the modalities of text, video, and audio with a CLIP adapter, while InstructAvatar utilized two separate adapters for emotion and expression description. EAT (Gan et al., 2023), on the other hand, leverages CLIP's text-image alignment capabilities, minimizing the CLIP loss between generated images and text descriptions to achieve zero-shot expression editing. Although AU-based text descriptions provide more detailed control over facial movements, this approach is computationally inefficient compared to directly encoding AUs. Additionally, the semantic space of AU-based text differs significantly from that of natural language, limiting the controlling ability. In conclusion, even though the aforementioned methods model the fine-grained condition with AUs, there are still limitations: 1) **Usage of AUs**: Previous methods primarily regarded AUs as a refinement of the emotion label, failing to achieve spatiotemporal fine-grained control. 2) **Language control**: Previous methods that utilized CLIP to encode AU-based text descriptions not only lost computational efficiency but also failed to ensure control capability over natural language.

## 3 CAFE-TALK

**Overview.** The fine-grained condition encodes subtle facial muscle movements intertwined with speech audio and the coarse-grained emotional label. Consequently, the lip movements could also be influenced by the fine-grained guidance instead of speech audio if we directly train the generative model conditioned on the three entangled conditions. Thus, we propose a coarse-to-fine pipeline with a two-stage training strategy – a base model with coarse-grained conditional control is trained in stage 1 with a similar structure validated in (Zhao et al., 2024; Ng et al., 2024), and the fine-grained condition is added to the base model with an adapter in stage 2. We denote the speech audio as $A$, coarse-grained condition $C$, and fine-grained condition as $F$. Concretely, the coarse-grained condition consists of the talking style $C_{ts}$, emotion label $C_{emo}$, and emotion intensity $C_{int}$. Meanwhile, the fine-grained condition is formulated as $n$ triplets with expression description $F_d$, the start and end frame index timestamp $F_s$, $F_e$, as:

$$F = \left\{ (F_d^0, F_s^0, F_e^0), (F_d^1, F_s^1, F_e^1), \cdots, (F_d^n, F_s^n, F_e^n) \right\}, \tag{1}$$

where the expression description $F_d$ is represented by binary AUs, where activated AUs guide the corresponding muscles engaged to generate local facial movements. The generator is $\mathcal{G}$, while the facial movement is noted as $M^{0:T}$ with $T$ frames.

### 3.1 BASE MODEL WITH COARSE-GRAINED CONTROL

In stage 1, we intend to train a base model that generates accurate lip movement with multimodal coarse-grained control, conditioning on the talking style and emotion status referred to facial images or emotional text descriptions. The base model, illustrated in Fig. 2, consists of stacked diffusion transformer blocks, integrating the encoded speech audio $A_f$ and coarse-grained condition $C_f$ with attention layers.

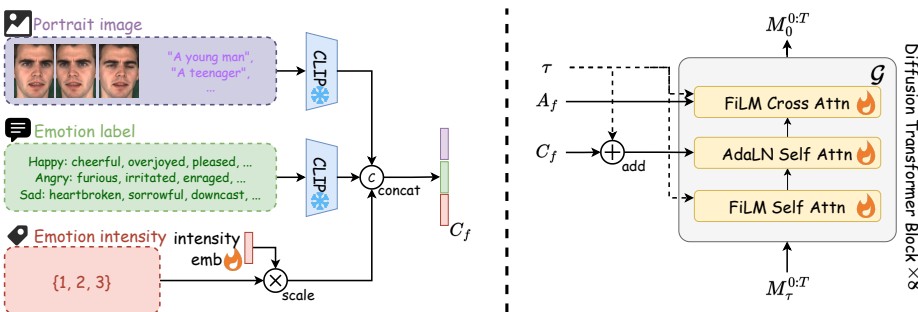

Figure 2: Stage 1 pipeline: a base model under transformer-diffusion structure is proposed to enable coarse-grained conditional control of temporal-consistent talking style, emotion status, and intensity.

### 3.1.1 MODEL STRUCTURE

**Coarse-grained condition.** In this work, we define the talking style, emotional status, and intensity as coarse-grained conditions that control the facial movement temporal-consistently in a multimodal fashion. Unlike existing methods (Fan et al., 2022; Daněček et al., 2023) utilized one-hot embedding for talking styles and emotion labels, Cafe-Talk takes CLIP embeddings as coarse-grained condition inputs. Inspired by Oh et al. (2019) inferring the speakers' appearances from speech audio, we obtain the talking style from the facial images or appearance attributes of the speakers. Meanwhile, the emotion label is encoded by the CLIP text encoder, which is augmented by synonyms (Xu et al., 2023; Ma et al., 2023). We learn a trainable embedding for the emotion intensity, which is scaled by a numerical emotion intensity factor, guiding the model to learn the ordinal nature of the intensity. The coarse-grained condition feature $C_f$ is the linear mapping result of the concatenation of talking style embedding, emotion embedding, and the emotion intensity on the feature dimension. Then, we merge the coarse-grained condition into the transformer diffusion block (Peebles & Xie, 2023) with a self-attention and feed-forward layer for efficient global conditional control.

**Speech audio condition.** We utilize a fixed pre-trained audio encoder (Conneau et al., 2020) to encode the speech audio, and then the acoustic feature is interpolated to 25fps to match the facial movement. We merge the acoustic feature $A_f$ with FiLM-based cross-attention layer (Perez et al., 2018), where an attention mask $Z_{align}$ is introduced within the cross-attention layer for lip synchronization, where the elements on and adjacent to the diagonal are $0$, and all other elements are $-\infty$.

Moreover, a self-attention layer with FiLM is also included, which is formed similarly to the speech audio, and the diffusion transformer block is constructed as stacked aforementioned layers and repeated 8 times in the generator $\mathcal{G}$. We list detailed hyperparameters in the App. C.1.

### 3.1.2 DENOISING PROCESS

Following the DDPM (Ho et al., 2020) definition, the forward noising process is defined as:

$$q\left(M_\tau^{0:T} \mid M_{\tau-1}^{0:T}\right) \sim \mathcal{N}\left(\sqrt{\alpha_\tau}M_{\tau-1}^{0:T}, (1-\alpha_\tau)I\right), \tag{2}$$

where $M_0^{0:T}$ denotes the predicted noise-free facial representation sequence, $\tau \in [1, \cdots, \dot{T}]$ denotes the forward diffusion step, and $\alpha_\tau \in (0, 1)$ follows a monotonically decreasing noise schedule such that as $\tau$ approaches $\dot{T}$, $M_\tau^{0:T}$ is sampled from the normal distribution.

We follow Ng et al. (2024) that predict $M_0^{0:T}$ directly from $M_\tau^{0:T}$, and the next step $M_{\tau-1}^{0:T}$ of the reverse process can then be obtained by applying the forward process to the predicted $M_0^{0:T}$. The prediction process can be mathematically formulated as Eq. 3:

$$\hat{M}_0^{0:T} = \mathcal{G}(M_\tau^{0:T}; \tau, A, C). \tag{3}$$

For the training objective, we adopt the simple loss (Ho et al., 2020)(Eq. 4). The speech audio and coarse-grained conditions are independently masked with a probability of $20\%$ during training for the CFG technique (Ho & Salimans, 2021), which enhances the controllability (shown in App. B).

$$\mathcal{L}_{simple} = ||M_0^{1:T} - \hat{M}_0^{1:T}||_2^2. \tag{4}$$

### 3.2 ADDING FINE-GRAINED CONDITIONAL CONTROL

**Fine-grained control adapter.** Natural facial expressions controlled by localized muscle movements should be generated automatically, instead of relying on hand-crafted post-processing. Thus, we integrate the fine-grained conditions into the base model to enable detailed facial control in the facial movement generation process. We choose binary AU sequences as the fine-grained conditions due to their ability to describe spatiotemporal localized movements and multimodal scalability. In our design, the activated AUs guide the excitation of the generated facial muscles rather than maintaining activation within a controlled interval. In order to add fine-grained control to the base model, we devise an adapter with the structure of FiLM self-attention layer (in Fig. 3) as extra network layers, which is optimized only in stage 2 while keeping the base model fixed. Additionally, a zero convolution layer (Zhang et al., 2023a; Wang et al., 2024) is added after the FiLM attention layer, which is initialized with zeros and is gradually optimized during training, ensuring a stable training process.

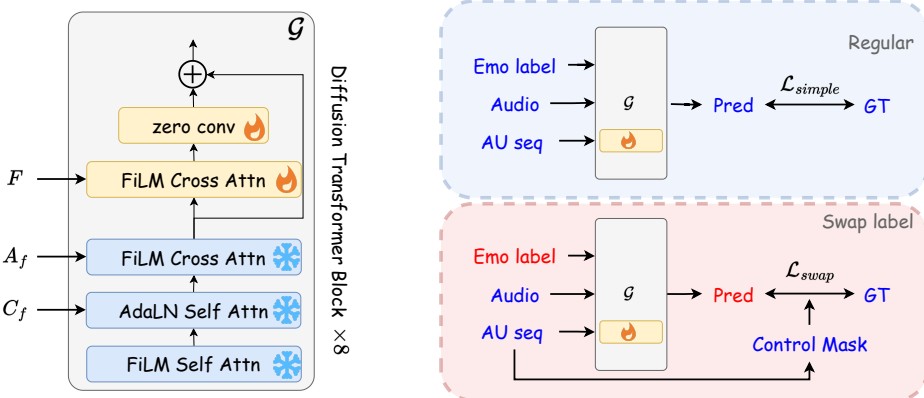

Figure 3: Stage 2 training pipeline: A fine-grained control adapter is inserted into the fixed base model and optimized in stage 2 with shuffling regular paired data or swapped emotion label.

**Swap-label mechanism.** The entanglement of the coarse- and fine-grained control cannot be completely eliminated only by network design because 1) **Entangled supervision**: the model is only trained on paired data $(A, C, F, M^{0:T})$, where coarse- and fine-grained conditions are entangled, therefore failing to generate, for instance, an angry facial movement with a sudden smile since no such data exists in the dataset; 2) **Insufficient training**: since the adapter is trained on the same emotion-annotated dataset as the base model, which generates a facial movement $\hat{M}_0^{0:T}$ that is similar to the ground truth $M^{0:T}$, the AU sequence may be treated as a condition for refinement rather than control; 3) **Impractical modeling**: users tend to apply fine-grained control over specific muscles within designated time intervals in the practical application, instead of the AU sequence with complete spatiotemporal movement information provided by the annotated training dataset. This inconsistency between training and inference can lead to suboptimal performance. To address the challenges above, we propose a swap-label training mechanism to break the provided coarse- and fine-grained condition pair and ensure the dominance of fine-grained control instead of the coarse-grained conditions. We visualize the swap-label mechanism in Fig. 3. During training, we replace the coarse-grained emotion label and obtain a swapped pair of data $(A, C', F, M^{0:T})$ while keeping the original AU sequence as fine-grained condition $F$ and the ground truth $M^{0:T}$. At the beginning of the training process, the generator initially produces $\hat{M}_0^{',0:T}$ according to the swapped emotion label $C'$. In order to disentangle the paired coarse- and fine-grained conditions and emphasize the controllability of the fine-grained condition, we optimize the adapter so that $\hat{M}_0^{',0:T}$ is as close as possible to $M_0^{0:T}$ when $F$ is given. The loss function is as follows:

$$\mathcal{L}_{swap} = ||Z_{ctrl}^{0:T} \odot M^{0:T} - Z_{ctrl}^{0:T} \odot \hat{M}_0^{',0:T}||, \tag{5}$$

where $Z_{ctrl}^{0:T}$ is a spatiotemporal mask representing the activated AUs within the fine-grained condition, and $\odot$ denotes the element-wise product. The swap-label mechanism can disentangle the coarse- and fine-grained conditions by sufficiently optimizing the adapter with swapped training pairs. While keeping the base model fixed, the swap-label mechanism consistently supervises the adapter to learn facial movement controlled by the AU sequences in the non-masked region. Meanwhile, the swap-label mechanism is randomly replaced with the regular simple loss (as Eq. 4), preserving accurate facial movements in the non-fine-grained controlled region. To effectively simulate the fine-grained conditions users provide in practical applications, the AU sequence requires sparsification in both temporal and spatial dimensions. Considering the temporal continuity of the conditions, we randomly drop elements triplets $(F_d, F_s, F_e)$ with a probability of $80\%$ from the AU sequence $F$. Within a specific remaining triplet, each activated AU is discarded with probability, preventing the suppression of the corresponding facial movements of non-activated AUs.

**Mask-based classifier-free guidance technique.** The influence of the fine-grained condition could spill over to the neighbor non-conditional temporal intervals due to the continuity of facial movements, as illustrated in the fourth row of Fig. 4. This issue cannot be rectified through training alone. Hence, we address it by employing a mask-based CFG technique based on Ho & Salimans

(2021) during inference, as:

$$\hat{M}_0^{0:T} = \mathcal{G}_\Phi + \alpha(\mathcal{G}_{A,C} - \mathcal{G}_\Phi) + \beta Z_{cfg}^{0:T} \odot (\mathcal{G}_{A,C,F} - \mathcal{G}_\Phi), \qquad (6)$$

where we abbreviate the notation of $\mathcal{G}(M_t^{0:T}; t, \cdot)$ as $\mathcal{G}_{\cdot\cdot}$. $Z_{cfg}^{0:T}$ is a temporal mask that marks the frames with user-provided fine-grained conditions. This temporal guidance prevents the leaking issue and performs seamless transitions under the denoising process. The $\alpha$ and $\beta$ are the guidance scale, and we leverage $\beta$ as the intensity control of the fine-grained condition shown in App. B.

**Text-based AU detector.** The choice of using AUs as the fine-grained condition facilitates multi-modal control, including facial image and text-based descriptions. Although we can employ off-the-shelf AU detectors (Baltrušaitis et al., 2016) from the facial portraits as the fine-grained condition, detecting AUs from textual descriptions of emotions or expressions remains an unexplored and challenging endeavor due to insufficient application. In order to train a text-based AU detector, we collect a dataset comprising text description-AU pairs from two data sources: on the one hand, we utilized the knowledge of LLM to obtain a limited number of high-quality emotions and expressions, along with their corresponding activated AUs. On the other hand, we supplement the dataset with a substantial number of emotion-AU pairs sourced from an image-based expression dataset (Mollahosseini et al., 2017). The lightweight detector employs CLIP-Adapter (Gao et al., 2024) as the backbone, considering CLIP's sensitivity to emotions and expressions when aligning textual and visual modalities. After passing through the adapter, the textual CLIP features are fed into the detector head to predict the activated AUs. Additionally, we use AU-based text descriptions for alignment to enhance generalization capabilities. We train our detector with a binary cross-entropy loss, while an InfoNCE loss is also included to mitigate the gap between AU and natural language descriptions. More technical details are listed in the App. E.

## 4 EXPERIMENTS

### 4.1 DATASETS

We use the public-available emotional talking face datasets MEAD (Wang et al., 2020) and RAVDESS (Livingstone & Russo, 2018) to train our model. MEAD captures approximately 40 hours of 48 participants' talking face clips in RGB video. It covers neutral and seven expression categories (angry, contempt, disgusted, fear, happy, sad, and surprised) in 3 intensity levels. Each participant is required to record 30 selected sentences for each expression type and intensity. RAVDESS contains 24 speakers and features over 6.5 hours of data, including 8 emotion categories (neutral, calm, happy, sad, angry, fearful, disgusted, and surprised). We split two participants from MEAD for validation and test set each and split RAVDESS following Peng et al. (2023b). Moreover, to enhance the generalization of lips synchronization, we collect approximately 252 hours of multilingual speaking videos from the internet to obtain a diverse range of speaking motions and obtain the ready-to-go dataset with a total duration of 157 hours after manually removing the segments with unsynchronized audio or occluded faces. We represent facial movements with a sequence of Apple ARKit blendshape coefficients[1], as each frame is a 51-dimension vector, and each dimension has specific facial muscle movements corresponding to it. An in-the-house video-based motion-capture model is utilized to obtain the facial movement. For semantic consistency, the AU sequence for each clip is obtained from the facial movements with a handcrafted rule (App. C.2).

### 4.2 EVALUATIONS

#### 4.2.1 METRICS

In our experiment, we adopt several metrics for comparison considering the following factors: 1) lip synchronicity, 2) emotion expressiveness and expression diversity, and 3) fine-grained condition control ability. Hence, we adopt lips vertices error (LVE↓, Richard et al. (2021)) and SyncNet score (SyncD↓ and SyncC↑, Chung & Zisserman (2017)) for lip synchronicity, and Acc ↑ and Div ↑ for expression. The calculation method for the metrics is detailed in App. D. Notably, we propose **Control Rate** (CR ↑) to evaluate the fine-grained control ability on the AU level, which is applied

---

[1]https://arkit-face-blendshapes.com/

on a fine-grained condition $(F_d, F_s, F_e)$ and a generated facial movements counterpart $M^{0:T}$, where $F_d$ contains $K$ activated AUs. For the $k$-th activated AU, the $\text{CR}_k$ is computed as the gap between the maximum coefficients in the controlled slot and the average of its non-controlled neighbors (5 frames, for 0.2s is the minimum duration of a noticeable expression (Ekman & Friesen, 1969)) as:

$$\text{CR}_k = \max(B_k^{F_s^i:F_e^i}) - \text{avg}(B_k^{F_s^i-5:F_s^i}, B_k^{F_e^i:F_e^i+5}), \tag{7}$$

where $B_k^{0:T}$ is the blendshape coefficient sequence from $M^{0:T}$ corresponding to the $k$-th activated AU. The larger CR value indicates more obvious and intense controlling effects.

### 4.2.2 COMPARISON WITH EXISTING METHODS

Since no previous methods exist on 3D talking faces with fine-grained spacial-temporal expression control, we individually train the base model on the MEAD and RAVDESS datasets and compare these methods on the benchmarks for lip synchronicity. We undertake further comparisons with the methods that utilize ARKit as the output. We report the emotion accuracy and diversity on the MEAD test set and the SyncNet score on a new benchmark with randomly selected audios from the LibriSpeech dataset (Panayotov et al., 2015). As shown in Tab. 1, our method outperforms the existing methods on both lip synchronization and expression generation.

Table 1: **Comparison with SOTA methods**: our method outperforms the existing SOTA methods on both lip synchronization and expression generation.

| | Rep | Emo | MEAD | | | RAVDESS | LibriSpeech | |
| --- | --- | --- | --- | --- | --- | --- | --- | --- |
| | | | LVE ↓ | Acc ↑ | Div ↑ | LVE ↓ | SyncD ↓ | SyncC ↑ |
| FaceFormer | Disp | ✘ | 16.36 | - | - | 9.24 | - | - |
| TalkSHOW | Coeff | ✘ | 13.21 | - | - | 12.54 | - | - |
| EMOTE | Coeff | ✔ | 9.37 | - | - | 14.67 | - | - |
| EmoTalk | ARKit | ✔ | 7.90 | 9.97% | 62.02 | 4.31 | 11.09 | 3.30 |
| UniTalker | ARKit | ✘ | 10.14 | 12.44% | 12.64 | 4.93 | **9.70** | 4.72 |
| Ours | ARKit | ✔ | **7.21** | **59.48%** | **119.36** | **3.93** | 9.76 | **5.55** |

Similarly, we evaluate the SyncNet score on the MEAD and RAVDESS test sets in Tab. 6 (App. D.3) demonstrating that our model achieves state-of-the-art performance in lip synchronization. These statistics highlight the structure of the diffusion model, which significantly enhances the diversity of output compared to discriminator-based models while avoiding the over-smoothing issue, and the use of AdaLN for coarse-grained condition injection, which effectively injects and maintains semantic control.

### 4.2.3 ABLATION STUDY

**Coarse-grained condition modeling.** To justify our choice of network design in coarse-grained condition modeling, we replace the AdaLN self-attention layer with a FiLM cross-attention layer used to incorporate fine-grained and speech audio conditions. As reported in Tab. 2, the base model with the AdaLN self-attention layer generates more appropriate and diverse expressions, indicating that our design is more efficient in modeling global conditions.

**Fine-grained design.** We conduct ablation studies on fine-grained controlling from the perspectives of model structure, training, and inference, as 1) removing the two-stage pipeline as the coarse- and fine-grained control are trained jointly, 2) removing the swap-label mechanism with only regular diffusion loss and 3) replacing the mask-based CFG technique with the one proposed by Ho & Salimans (2021). We collect a test set with in-the-wild speech audios with single or multiple AUs activated and keep the random seed fixed. As shown in Tab. 3 and Fig. 4, we conclude that 1) **Without the two-stage pipeline**, the jointly trained model exhibits fine-grained control, but its lip movements are significantly unsynchronized. 2) **Without the swap-label mechanism**, the adapter trained with paired conditions fails to generalize on conflict conditions, resulting in invalid fine-grained control and unsynchronized lip movements. 3) **Without the mask-based CFG technique**, the generation encounters issues of temporal leakage, making the fine-grained control difficult to observe within

the control interval. We observe that the SyncNet scores degrade when fine-grained control is incorporated, compared to the counterpart without fine-grained control (in Tab. 1). To further investigate this, we analyze the contribution of each AU to the evaluation result, as presented in Tab. 4. The analysis reveals that inappropriate activations of lower-face AUs disproportionately influence facial movements, resulting in suboptimal lip synchronization. This explains why the SyncNet score degrades more without the mask-based CFG technique, as the inappropriate fine-grained condition leaks and contaminates facial movements.

Table 2: Coarse-grained design ablation

|  | LVE ↓ | Acc ↑ | Div ↑ |
|---|---|---|---|
| Ours (FiLM) | 7.64 | 48.19% | 114.26 |
| Ours (AdaLN) | **7.21** | **59.48%** | **119.36** |

Table 3: Fine-grained design ablation

| Method | CR ↑ | SyncD ↓ | SyncC ↑ |
|---|---|---|---|
| w/o two-stage | 0.35 | 11.94 | 2.78 |
| w/o swap-label | 0.12 | 11.02 | 4.07 |
| w/o masked CFG | 0.27 | 10.66 | 4.74 |
| Ours | **0.51** | **10.14** | **5.38** |

Table 4: SyncNet scores on single-AU fine-grained control: with fixed speech audio and coarse-grained conditions, the results controlled by upper-face AUs do not suffer from lip-sync issues (+0.156 on average), whereas those with lower-face AUs exhibit unsynced lips (+0.693).

| upper | AU | Null | AU01 | AU02 | AU04 | AU05 | AU06 | AU07 | AU45 |
|---|---|---|---|---|---|---|---|---|---|
|  | SyncD | 9.667 | 9.967 | 10.365 | 9.784 | 9.735 | 9.982 | 9.613 | 9.615 |

| lower | AU | AU09 | AU10 | AU12 | AU14 | AU15 | AU17 | AU20 | AU26 | AU28 |
|---|---|---|---|---|---|---|---|---|---|---|
|  | SyncD | 10.251 | 10.465 | 10.788 | 10.194 | 10.354 | 10.614 | 10.090 | 10.423 | 10.066 |

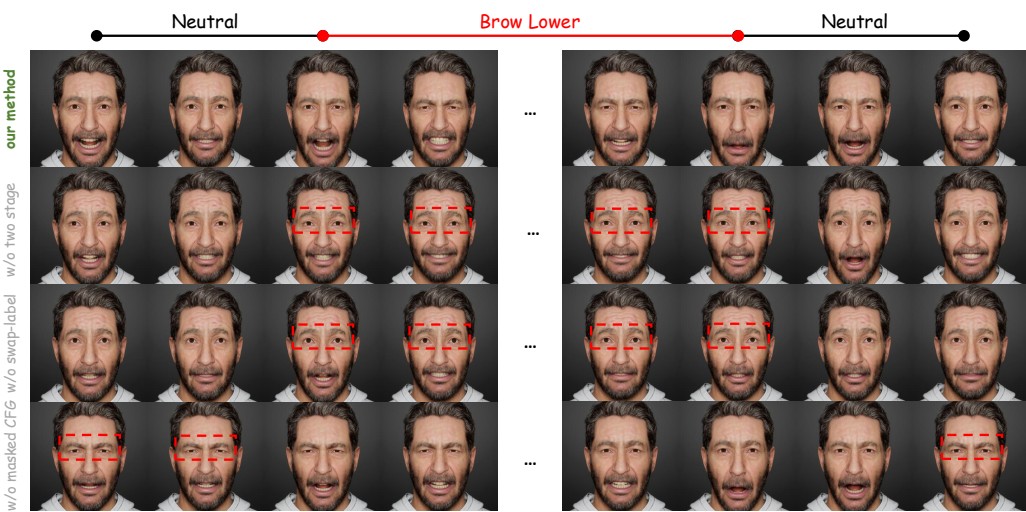

Figure 4: Ablation on fine-grained design: we illustrate the ablation by generating facial fine-grained condition *Brow Lower* within a neutral talking animation.

We visualize the training loss in Fig. 5, and conclude that 1) Incorporating the fine-grained condition specifies the generation target, and simplifies the training objective (*base vs. w/o two-stage* and *base vs. w/o swap*), and 2) Swap-label mechanism complicates the training objective with conflict conditions, as the larger loss is observed, enabling the adapter to dominate fine-grained control.

### 4.2.4 USER STUDY

We conduct a user study involving 38 participants who are first required to select from EmoTalk, UniTalker, and our method the animation they deem most natural, expressive, and exhibiting the best lip-syncing. Our model achieves the highest performance across metrics as shown in Fig. 6.

Then, the participants are invited to evaluate the fine-grained control ability by comparing it with the animation without the fine-grained control. In this evaluation, the fine-grained control capability, as well as the overall performance is widely accepted performance by participants. We present coarse- and fine-grained control in Fig. 7.

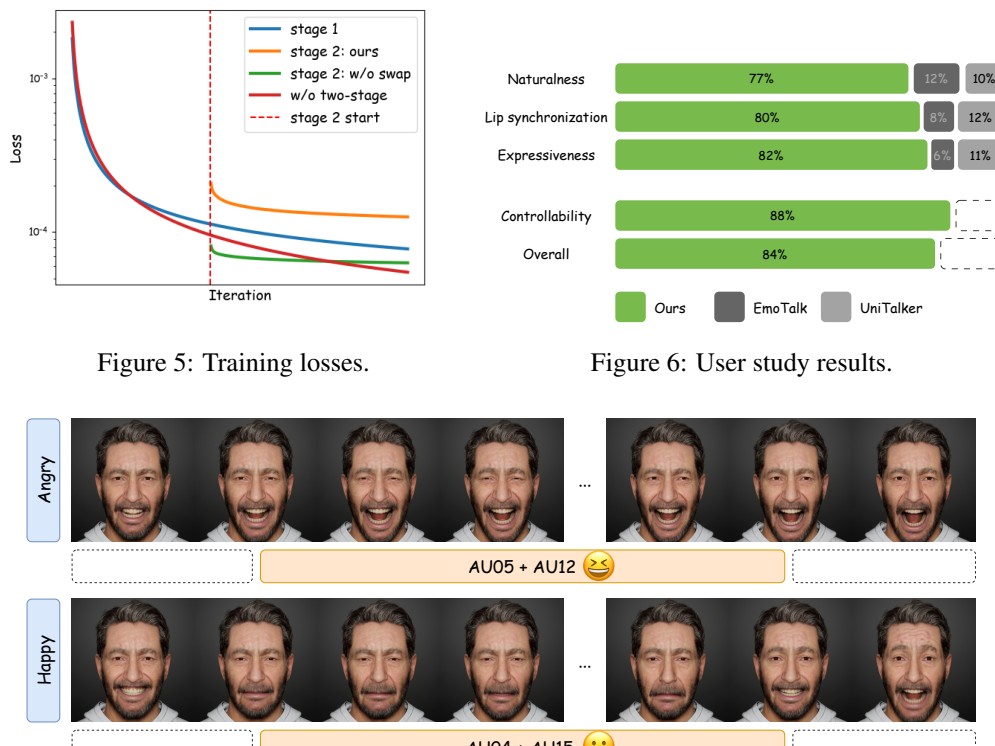

Figure 5: Training losses.    Figure 6: User study results.

Figure 7: Cafe-Talk generates natural movements with conflict coarse- and fine-grained conditions.

## 5 CONCLUSION

We present Cafe-Talk, the first diffusion-based 3D talking face model featuring both coarse- and fine-grained multimodal conditional control. The fine-grained condition is represented with AUs, enabling localized and flexible control. Cafe-Talk is designed to add a fine-grained control adapter into a diffusion-based talking face model with coarse-grained control. Provided with conflict conditions during training and masked temporally during inference, the fine-grained condition dominates the generation precisely. We extend the natural language control by a detector aligned with AUs. Cafe-Talk has achieved superior performance across all evaluations, and fine-grained control has received wide acceptance in user studies, validating the efficacy of the design. However, our model still has limitations that need to be overcome. The failure cases show that there could be inaccurate lip movements when the fine-grained conditions of the lower-face AUs explicitly conflict with speech audio. Our near-term research efforts are geared towards immediate improvement with a better solution to harmonize the fine-grained control of the lower face with speech audio. Meanwhile, other aspects of 3D talking faces and applications could be researched in the long term. For example, it would be of interest to researchers to explore more real-life AI agents with LLMs and flexible control offered by our Cafe-Talk.

### ACKNOWLEDGMENTS

This work was supported by the National Key R&D Program of China (2023YFF1203803, 2023YFC3604500), Beijing Science and Technology Plan Project Z231100005923039, and National Natural Science Foundation of China (62441201, L2324214).

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

## A  OVERVIEW

We form the appendix as follows: in App. B we present more visualization results with coarse- and fine-grained control. In App. C, we depict the implementation details of our model, and provide more details for evaluation in App. D. Finally, the text-based AU detector is described in App. E.

## B  MORE RESULTS

We demonstrate emotion control by the coarse-grained condition in Fig. 8. Since the emotion is encoded into the CLIP embedding space, our model has the capability of not only the in-domain emotion labels (*i.e., angry, fear, happy, sad and surprised*), but also the unseen emotion words (*e.g., shocked*). In addition, even though not adopted during training, the coarse-grained condition also supports facial portraits as condition input, thanks to the multimodality alignment of the CLIP.

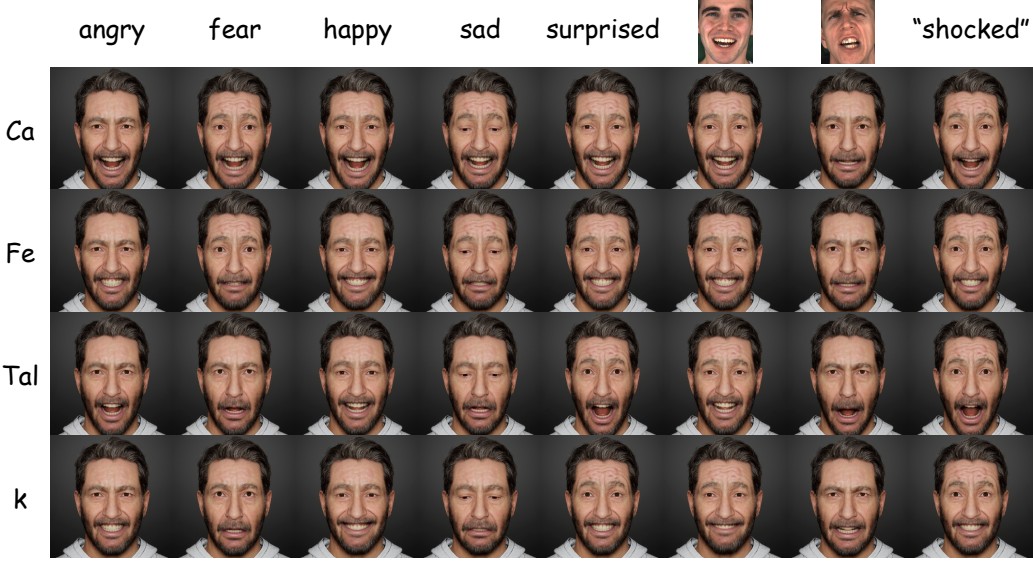

Figure 8: Results with coarse-grained control

However, the emotion perceived by CLIP does not fully ensure that the generated facial movements precisely match the intended expression. This underscores the importance of fine-grained conditional control. In Fig. 9, we present the results of our model generated with multimodal detected AUs.

We visualize the facial movement coefficients generated with different CFG scale $\beta$ in Fig. 10, where the fine-grained condition precisely occurs in its slot due to our mask-based CFG technique. Due to the intensity modeled by the CFG technique, the generated results exhibit different rhythms, which cannot be achieved by post-processing *e.g.*, linear multiplication, or summation of the coefficients.

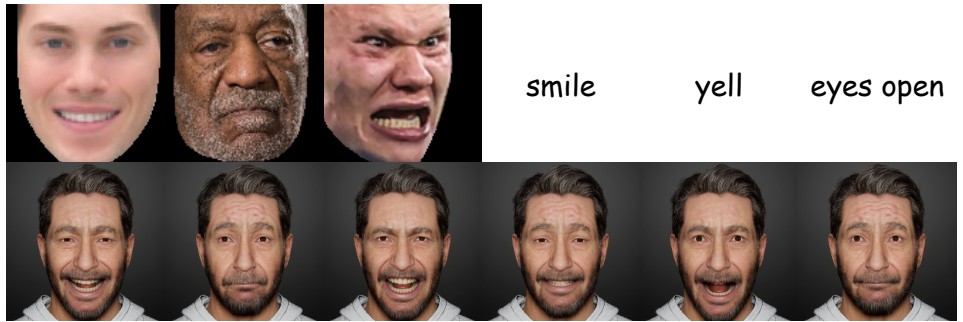

Figure 9: Results with fine-grained control

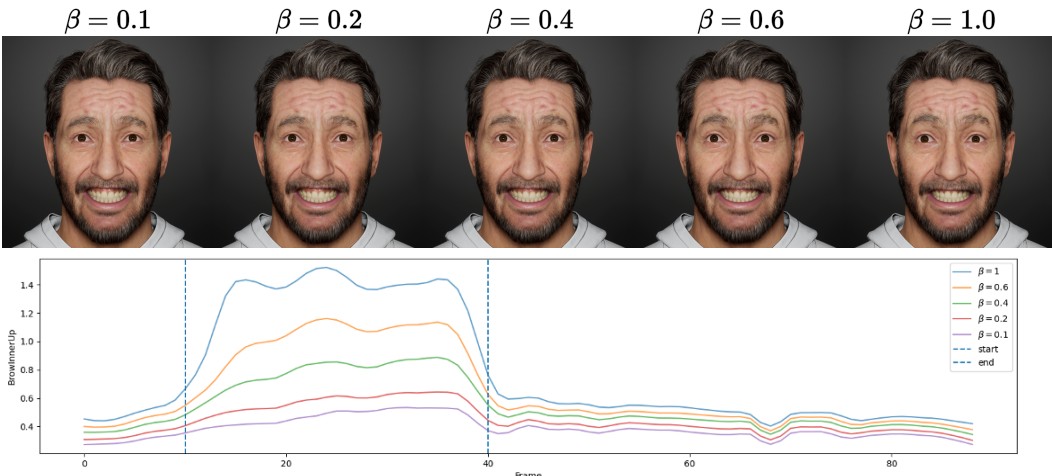

Figure 10: Controlling intensity with CFG scale: the eyebrows are raised higher with bigger $\beta$

## C  IMPLEMENTATION DETAILS

### C.1  DIFFUSION TRANSFORMER

In the diffusion transformer block, the attention layers take 512-dimensional features as input and are equipped with 8 heads. We utilize an AdamW optimizer with a learning rate of 0.0001, and both stages are trained with a batch size of 16 on 8 Nvidia V100 GPUs for 400k and 300k iterations ($\sim 4$ days each). More information are listed in Tab. 5

Table 5: Parameters and training strategies for modules in the diffusion transformer

| Module | Parameters (M) | Training strategy |
|---|---|---|
| Wav2Vec2-xslr-300m | 315.43 | Fixed |
| Base model | 25.09 | Opt. in stage 1, fixed in stage 2 |
| Adapter | 5.39 | Opt. in stage 2 w/ swap-label |
| Total | 345.91 | — |

### C.2  RULE-BASED AU DETECTION

Unlike existing methods utilizing an off-the-shelf detector to obtain AUs, we devise a rule-based algorithm (Alg. 1) to binarize from the coefficient sequence for multi-fold reasons: 1) Semantical mismatches exist between the AU detector prediction and the ground-truth facial movements, making it fail to learn the co-relationship (Fig. 11). Previous methods may not run into such difficulty

since only salient AUs within a sequence are selected. 2) Our algorithm provides flexibility: if only binarizing the facial movements with a threshold, the fine-grained condition will activate corresponding muscles for the entire interval, for we encourage the corresponding muscles to fluctuate between activation and deactivation under fine-grained control.

---

**Algorithm 1** Binarization Algorithm Process

---

**Input:** Expression sequence $M^{0:T}$
**Output:** AU condition sequence $F$

**Step 1:** Binarize $M^{0:T}$ by a threshold
**Step 2:** Apply max pooling with a random kernel size to the binarized sequence
**Step 3:** Merge adjacent active sequences randomly if the gap between them is less than a spacing threshold

---

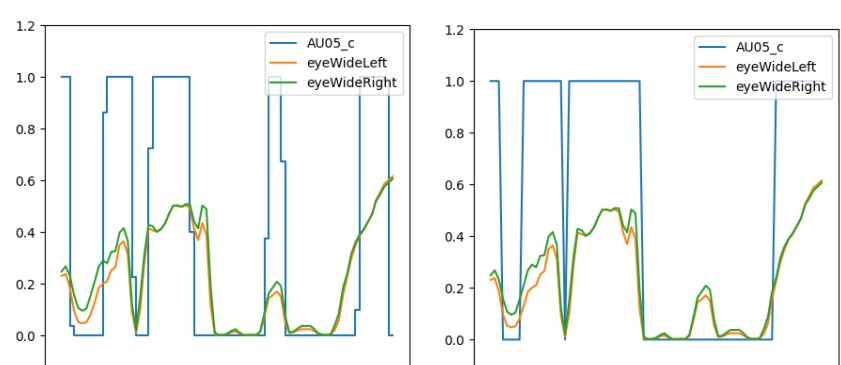

Figure 11: The AU sequence obtained by off-the-shelf detector (Baltrušaitis et al., 2016) (left) and our binarize algorithm (right). The rule-based algorithm produces fine-grained conditions with semantic consistency.

### C.3 MISCELLANEOUS

With a base model trained jointly on the MEAD and RAVDESS datasets, we utilize the 157-hour internet dataset in stage 2. Considering the unreliability of the concurrent video-based emotion recognition methods, we randomly arrange the emotion label and intensity as input, following the swap-label mechanism to enhance the fine-grained condition controllability. During training, we trim the audio up to 10 seconds for training efficiency.

## D EVALUATION

### D.1 METRICS

**Lips vertices error**    We utilize lips vertices error (LVE) (Richard et al., 2021) to evaluate the lips motion synchronization to the ground truth, which is defined as the maximal L2 error of all lip vertices on the frame level and reports the average over all frames in the testing set.

**SyncNet score**    To evaluate the lips synchronization without ground truth, we adopt the SyncNet (Chung & Zisserman, 2017) minimum distance (SyncD) and confidence score (SyncC), which is commonly adopted for evaluating lips synchronization on video-based talking-head and the videos are rendered by Unreal Engine MetaHuman [2] in our case.

---

[2] https://www.unrealengine.com/en-US/metahuman

**Emotion accuracy and diversity**     We test the expression of the generated facial movements with a transformer-based emotion classifier. The classifier first encodes the facial movement $M^{0:T}$ into feature $M_f^{0:T}$ and classifies it to the emotion class with a classifier head. We report the accuracy of the prediction and the feature-level standard deviation as diversity.

## D.2    METRICS OF ACC

**Transformer-based classifier**     We introduce a Transformer-based classifier to classify the emotion category. As the model structure is shown in Fig. 12, the classifier takes the facial movements with an additional `<cls>` token as the input and maps them into latent space with an MLP. Then, with positional embedding, it is fed into the Transformer block, which consists of one layer of the Transformer encoder layer with 8 heads. Finally, a classifier head takes the first token from the encoder and outputs the prediction. The classifier is trained on the MEAD dataset with cross-entropy loss. The training process lasts for 200 epochs, where the learning rate is set to 1e-4 and the batch size is set to 128. Notably, we randomly split the dataset into the training set, validation set, and testing set, with quantities of 70%, 15%, and 15%, to ensure the in-domain classifying ability. The classifier achieves accuracies of 88.28% on emotion classification.

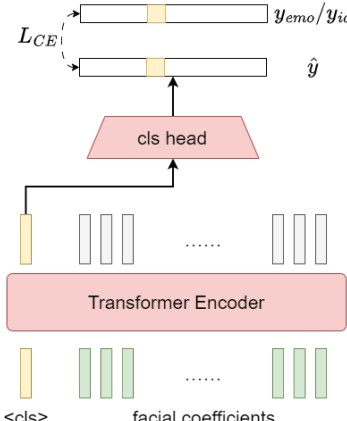

Figure 12: Model structure: the transformer-based classifier

## D.3    PERFORMANCE

**Comparison with existing methods.**     We report the SyncNet score on the MEAD and RAVDESS datasets in Tab. 6, further demonstrating the superiority of our model in lip synchronization.

Table 6: The SyncNet score on MEAD and RAVDESS test set.

| Models | MEAD | | RAVDESS | |
|---|---|---|---|---|
| | SyncD $\downarrow$ | SyncC $\uparrow$ | SyncD $\downarrow$ | SyncC $\uparrow$ |
| UniTalker | 11.52 | 3.05 | 9.57 | 2.35 |
| EmoTalk | 11.63 | 2.91 | 9.61 | 2.29 |
| Ours | **11.22** | **4.33** | **9.33** | **3.95** |

We also report the size of the relevant models and the inference time for 5-second audio on the Nvidia 2080Ti GPU in Tab 7. The inference latency is primarily caused by the sampling and denoising process of the current diffusion architecture, which could be partially alleviated with a more advanced GPU. Our method's high controllability, precise manipulation, and diverse outputs are compatible with flexible offline facial editing and time-tolerant AIGC talking animation generation in real-world applications.

Table 7: Size and inference time cost on related methods.

| Models | Num of parameters (M) | Inference time (sec) |
|---|---|---|
| FaceFormer | 94.7 | 3.34 |
| TalkSHOW | 94.9 | 3.69 |
| EMOTE | 109.6 | 3.73 |
| EmoTalk | 640.6 | 10.99 |
| UniTalker | 96.0 | 3.20 |
| Cafe-Talk | 345.9 | 14.71 |

**Ablation study on emotion label.** Since the base model is trained with paired emotion label and speech audio, where speech audio as well conveys emotion implicitly, we conduct a series of ablation studies to evaluate the controllability of the emotion label by 1) **masking** the emotion label within the CFG technique (Eq. 6) and, 2) **swapping** the emotion label randomly. Reported in Tab. 8, the emotion accuracy drops sharply without the correct emotion label, indicating the label serves as an explicit emotion condition, which the model prioritizes and learns as the dominant factor.

Table 8: Ablation study on the controllability of emotion label.

| | Acc ↑ | Div ↑ |
|---|---|---|
| Mask emotion label | 23.68% | 110.96 |
| Swap emotion label | 57.22%, label as GT
10.86%, audio as GT | 106.33 |
| Paired (reported in Tab. 1) | 59.68% | 119.36 |

**Ablation study on emotion intensity.** We evaluate how the emotion intensity impacts the generation and modeling design. As shown in Tab. 9, we first provide the detailed matrics on the MEAD test set within each intensity level. We observe that higher intensity levels result in higher Acc, indicating that incorporating expression intensity enhances the emotional expressiveness of the generated results. Then we re-generate the facial movements on the MEAD test set, and observe a sharp drop in both Acc and Div, demonstrating that emotion label control fails in this extreme case and confirming that our design enables the model to learn the ordinal relationship of intensity. Finally, in an ablation study removing intensity from the model pipeline, we observed a decline in both Acc and Div, suggesting the model is more likely to generate expressions with medium intensity. This further highlights the necessity of intensity for flexible control.

Table 9: Ablation study on coarse-grained intensity condition.

| | Acc ↑ | Div ↑ |
|---|---|---|
| intensity = 1 subset | 51.68% | 111.16 |
| intensity = 2 subset | 63.86% | 119.75 |
| intensity = 3 subset | 66.26% | 119.48 |
| intensity = 0 | 19.07% | 67.09 |
| remove intensity | 53.27% | 115.17 |
| Ours (reported in Tab. 1) | 59.48% | 119.36 |

**Ablation study on coarse-grained design.** We illustrate the emotion accuracy with the confusion matrix in Fig. 13, corresponding to Tab. 2. We also notice that the major mispredicted pairs are "disgusted-angry" and "surprised-fear", which is acceptable since the definition boundary is ambiguous.

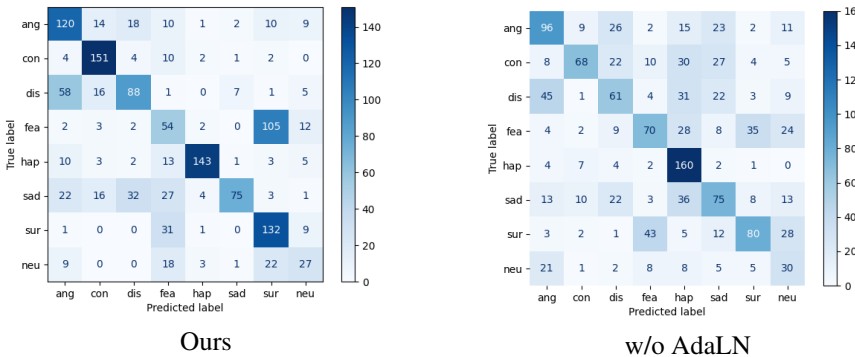

Ours             w/o AdaLN

Figure 13: Confusion matrix: coarse-grained design with AdaLN achieves efficient control.

## E  TEXT-BASED AU DETECTOR

**Model structure**  We construct a text-based AU detector based on CLIP (Radford et al., 2021) for its multimodal alignment ability. As shown in Fig. 14, the text input is first encoded by the fixed CLIP encoder, then fed into an adapter (Gao et al., 2024) with normalization. Finally, a classification head is utilized to detect the activated AUs. Concretely, the CLIP adapter (Gao et al., 2024) is a bottleneck 2-layer MLP ($[512 \rightarrow 128 \rightarrow 512]$) with residual connection, activated by ReLU activation function, and the parameters are counted in Tab. 10.

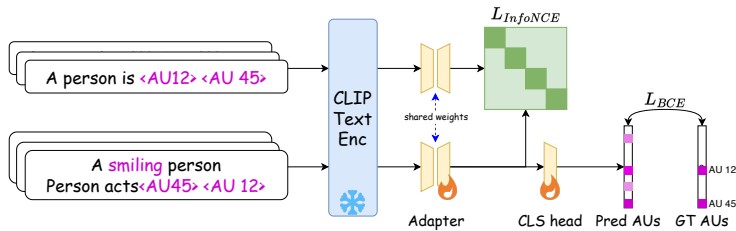

Figure 14: Textual AU detector pipeline

Table 10: Parameters and training strategies for textual AU detector

| Module | Paramters (M) | Training strategy |
|---|---|---|
| CLIP text encoder (ViT-B/32) | 37.82 | Fixed |
| CLIP adapter + head | 0.13 | Opt with InfoNCE and BCE loss |
| Total | 37.95 | — |

**Dataset**  We leverage the powerful language analysis capabilities of GPT to enhance our model's ability to perform expression analysis on an open vocabulary. The prompt used is as follows:

> Please generate a series of complex and abstract facial expressions, each involving specific Action Units (AUs) from the following list: [AU key-value pair, abbreviated, *e.g.*, AU01:InnerBrowRaiser]. Expressions should be a mix of both simple (e.g., frown, raised eyebrows, kiss, wink) and complex combinations (e.g., skeptical with one eyebrow raised, incredulous with raised brows and open mouth). Additionally, include mixed emotions (e.g., happy-surprise, sad-relief) with the corresponding AUs. Provide at least 50 different expressions, ensuring a wide variety of emotional and physical expressions. Output the results as a Python

dictionary with each expression labeled as either an emotion or an expression, along with the corresponding AUs.

We obtained 228 text-AU pairs in total, which cover both emotion and expression, examples are demonstrated in Tab. 11. Then we formulate the generated corpus with the template shown in Tab. 12, and obtain the natural facial descriptions, *e.g.*, "A shocked person." Meanwhile, we enrich the dataset by collecting the emotion-AU pairs from AffectNet (Mollahosseini et al., 2017), where the emotion is manually annotated and augmented with synonyms, and the AUs are detected by an off-the-shelf detector (Baltrušaitis et al., 2016).

Table 11: Examples of text-AU pair generated by GPT

| Description | Type | AUs |
|---|---|---|
| Annoyed | Emotion | AU04, AU07, AU23 |
| Shocked | Emotion | AU01, AU02, AU05, AU26 |
| Anger-Disgust | Emotion | AU04, AU09, AU10, AU17 |
| Pain with Squinting | Expression | AU04, AU07, AU10, AU23 |
| Happiness with Broad Smile | Expression | AU06, AU12, AU25 |
| Frowning | Expression | AU04, AU15 |

Table 12: The templates designed to formulate the natural language descriptions

| Description templates | |
|---|---|
| A [desc] person | A woman acts like [desc] |
| A person looks [desc] | A person is [desc] |
| A person seems [desc] | A person behaves [desc] |
| A man acts like [desc] | A person is talking with [desc] |

**Training** The model is trained with 200 epochs, where the batch size is set to 128. We utilize the Adam optimizer and the learning rate is set to 0.0001.

**Evaluation** The CLIP adapter and the classification head are evaluated on the textual AU detection task and achieve an F1-score of 0.92 and 0.98 on the testing set (in-domain template), with and without the InfoNCE loss, respectively. Moreover, we visualize the CLIP embeddings $f_{clip}$ and the adapted embeddings $f_{adaptive}$ with and without the InfoNCE loss in Fig. 15 to evaluate the adapter. In conclusion, with a contrastive learning strategy, the adapter erases the domain gap between AU and natural language descriptions, while preserving the AU information. Meanwhile, in Tab. 13, we ablate the importance of the alignment and knowledge of GPT, with which the detector shows better generalization.

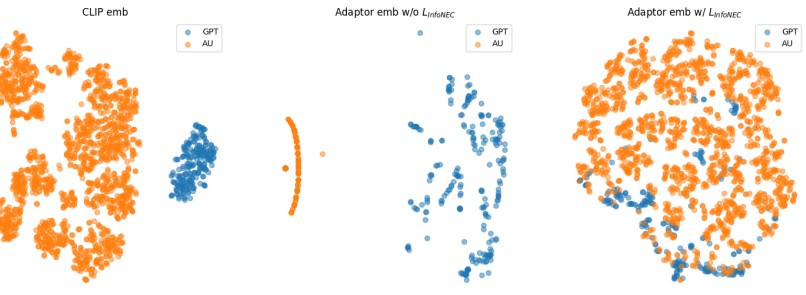

Figure 15: The T-SNE visualization of the CLIP and adapted features.

Table 13: Ablation study: we ablate the detector by 1) removing the contrastive learning (as *w/o alignment*), and 2) training without the GPT generated pairs. (as *w/o GPT pairs*). AUs in red denote incorrect predictions.

| Description | ours | w/o alignment | w/o GPT pairs |
|---|---|---|---|
| A person is smiling | AU12 | AU12 | AU05 AU10 AU12 |
| A person is crying | AU04 AU15 | AU04 AU15 | AU04 AU07 |
| A person is blinking | AU45 | AU45 | AU45 |
| A yawning person | AU26 | AU26 | AU04 AU10 |
| A person closes his eyes | AU45 | AU20 | AU05 AU10 AU04 |
| A not happy person | AU04 AU15 | AU04 | AU01 |

