# OpenReview forum: "Cafe-Talk: Generating 3D Talking Face Animation with Multimodal Coarse- and Fine-grained Control"
_ICLR.cc/2025/Conference — ICLR 2025 Poster_

### Official Review · Reviewer_UmXQ · 2024-10-31

**Soundness:** 2
**Presentation:** 2
**Contribution:** 2
**Rating:** 6
**Confidence:** 2

**Summary:**

This paper proposes CAFE-TALK, a 3D talking face generation model enabling coarse- and fine-grained multimodal conditional control. A two-stage training pipeline is designed to incorporate coarse control and fine-grained control. To further disentangle coarse- and fine-grained conditions, a swap-label training mechanism and mask-based CFG technique are introduced. In addition, a text-based AU detector is devised to extract AU fine-grained conditions from user-input natural language descriptions.

**Strengths:**

1. CAFE-TALK's primary contribution lies in enabling fine-grained, multimodal conditional control over talking faces. This is achieved through a novel two-stage training pipeline incorporating a swap-label training mechanism and a mask-based CFG technique.
2. Extensive experiments and ablation studies demonstrate that the proposed CAFE-TALK model outperforms baseline methods in both lip synchronization and expression generation.
3. The paper is well-structured and easy to follow.

**Weaknesses:**

1. When fine-grained control is applied to talking faces, we notice abrupt shifts in expression, such as from happy to angry. Such changes can appear unnatural and make the work less appealing if no real-world application scenarios can be found.
2. The authors demonstrate the method using examples of a single individual, raising questions about its generalizability across different faces. Additionally, fine-grained text control is only shown in a limited number of scenarios. Could the authors provide more examples to showcase this control in diverse cases?
3. For facial muscle control, only the Brow Lowerer muscle is demonstrated. How effective is the model at controlling other facial muscles?
4. Beyond fine-grained text-based control, can CAFE-TALK achieve comparable fine-grained control using images as input?

**Questions:**

Questions are listed in the weakness part.

---

> ### Author Response · Authors · 2024-11-22
> **Response to Reviewer UmXQ**
>
> Thank you for your positive comments and constructive suggestions. We address all your concerns, and we shall provide more exemplars for better understanding to the readers.
>
> **W1**: Dramatic emotional changes have numerous applications in games and movies, such as reacting with panic to a sudden gunshot or laughing out loud upon realizing a prank. Notably, Cafe-Talk can control both abrupt and subtle expression changes, such as a slight and sudden smile in a neutral state. This fine-grained control enhances the naturalness of animation, which is challenging to achieve in previous works. We will include specific application scenarios in the introduction section of our camera-ready version to highlight the use cases of Cafe-Talk.
>
> **W2**: We shall add more examples to address your concerns. We utilize Apple ARKit as our facial movement representation, which is avatar-independent and muscle-based. Therefore, the generated results can be easily retargeted to another avatar, as shown in the uploaded demo video with another avatar (06:18-06:24) and *sup_video_1.mp4*. Meanwhile, we demonstrate the fine-grained text control with "laugh" (06:50-06:57), "smile and sob" (07:03-07:11) in the demo video, and we present more examples in the newly uploaded video *sup_video_2.mp4*.
>
> **W3**: As shown in Tab. 4, Cafe-Talk forms the fine-grained condition with 16 AUs. We also demonstrate the controls of blink (06:36-06:41), outer brow up (06:42-06:50), and combinations of multiple AUs.  We offer more examples in the newly uploaded video *sup_video_3.mp4*, demonstrating muscles including eyebrows, eyelids, and nose.
>
> **W4**: Indeed. Cafe-Talk can achieve fine-grained control with a reference image, by leveraging the off-the-shelf AU detectors, e.g. OpenFace, to predict activated AUs and their intensity as the fine-grained condition.

---

> > ### Comment · Reviewer_UmXQ · 2024-11-26
> > **Review Update: Response to Rebuttal**
> >
> > I would like to thank the authors for addressing my concerns. I tend to keep my original rating. I would also suggest that the authors provide more examples of different avatars. Currently, this is only limited to two avatar subjects.

---

> > > ### Author Response · Authors · 2024-11-26
> > > **Response to Reviewer UmXQ**
> > >
> > > Dear Reviewer UmXQ,
> > >
> > > We want to thank you for your suggestions. We have added more avatars to our newly uploaded video  `sup_video_4.mp4`, and will present more diverse avatars in the demo video of our future works.

---

### Official Review · Reviewer_MxhK · 2024-11-01

**Soundness:** 2
**Presentation:** 3
**Contribution:** 3
**Rating:** 6
**Confidence:** 5

**Summary:**

This paper proposes ***Cafe-Talk***, a 3D talking face generation model that excels in multimodal control, enabling both coarse and fine-grained manipulation of facial expressions.

**Strengths:**

**Ideas**:
***Cafe-Talk*** can handle both coarse and fine-grained control conditions and ensure the priority of fine-grained control signals during Stage II training and inference.

**Methodology**:
1. Unlike other methods, ***Cafe-Talk*** utilizes binary AU sequences as fine-grained conditions, avoiding the issues of poor controllability and low differentiation of facial details caused by using CLIP features of text prompts.

2. ***Cafe-Talk*** designed a swap-label mechanism to disentangle coarse and fine-grained control signals and analyses in sec. 3.2 is reasonable.

3. The mask-based CFG technique is another standout feature of ***Cafe-Talk***. It addresses the challenge of controlling the influence of fine-grained conditions on the generated facial movements, ensuring that the effects are localized to the intended time intervals. This technique prevents the leakage of fine-grained control into non-controlled intervals, which could lead to unnatural expressions.

The experiment is quite complete. Good img and charts.

**Weaknesses:**

Some details are unclear; see the "Questions" section below for more information.

**Questions:**

1. In Table 1, why not show the comparison results of syncnet metrics on the MEAD and RAVDESS datasets? On the other hand, there is little point in making comparisons on your own collected dataset. Instead, you need to conduct experiments in the absence of the 157-hour dataset.

2. Some details of the datasets are not fully elaborated. What is the tool for extracting Apple ARKit? The details of the "in-the-house video-based motion-capture model" in L361-362 need to be explained. Did you collect a video dataset using an iPhone depth camera and train a model on it to extract ARKit coeffs?

3. What is the mask $Z_{align}$ used for in L225? What does $i,j$ mean in eq. 2? And how to determine the $Z_{ctrl}^{0:T}$ in eq. 6? Was this derived manually based on FACS AU's and Apple ArKit's semantics?

4. The $CR$ metric defined in eq. 8 is not very reasonable. CR measures the difference in expression coeffs between the fine-grained control period and outside of it. However, it does not assess whether the fine-grained control matches the intended facial expressions accurately. It only indicates the presence of a change, not the correctness of that change relative to expected expressions. Modifying the CR metric or adding additional metrics to describe the accuracy of fine-grained modifications is recommended.

5. What are **Talking Styles** in coarse-grained controls, and what are they used for? In the demo video, talking style is expressed as gender, age, and race, and it is "A senior actor" in Fig. 1. However, ARKit is ID-independent, and talking style is not reflected as an input in the coarse-grained conditional control model in Fig. 2.

6. You have chosen ARKit as your motion representation, so why not use it but AU as the fine-grained control signals? By designing a text-to-ARKit model, you will obtain a fine-grained modifier sequence and can directly use it as a motion prior.

---

> ### Author Response · Authors · 2024-11-22
> **Response to Reviewer Mxhk (1/2)**
>
> Thanks for your positive comments and constructive suggestions. We will make clearer statements in our camera-ready version.
>
> **Q1**: We shall add the SyncNet results on the MEAD and RAVDESS benchmarks, as shown in the table below.
> | Models     |    MEAD SyncD $\downarrow$    |    MEAD SyncC $\uparrow$   |    RAVDESS SyncD $\downarrow$   |    RAVDESS SyncC $\uparrow$   |
> |------------|:-----------:|:-----------:|:-----------:|:-----------:|
> | UniTalker  |    11.52    |     3.05    |     9.57    |     2.35    |
> | EmoTalk    |    11.63    |     2.91    |     9.61    |     2.29    |
> | Ours       |    **11.22**    |     **4.33**    |     **9.33**    |     **3.95**    |
>
> We plan to make a more straightforward statement in the camera-ready version to address the concerns about the datasets. 1) The results of our model in Tab. 1 show the performance in the absence of the 157-hour dataset. The 157-hour dataset was only used for our swap-label mechanism of fine-grained conditions in stage 2 training, and we exclude this extra dataset in Tab. 1 for a fair comparison. 2) The "in-the-wild" test subset mentioned in Tab 1 is not derived from the 157-hour dataset but from 10 randomly selected segments from the LibriSpeech dataset. LibriSpeech is an audio-only dataset without a video counterpart, meaning that no model can be trained on this dataset. Therefore, it serves as a fair benchmark for testing the generalization ability.
>
> **Q2**: In fact, we collected a video datasets with ARKit coefficients by iPhone, and then trained a mobilenet. This is reproducible by replacing it with open-source Google Mediapipe.
>
> **Q3**: $Z_{align}$ is the attention mask applied on the cross-attention layer, similar to the alignment bias proposed in Faceformer [1], and is introduced for correctly aligning the audio-motion modalities. $Z_{align}^{i, j}$ represents the element in the $i$-th row and $j$-th column. $Z_{ctrl}$ implements the conversion from AUs to ARKit based on the semantics of FACS. We shall add detailed explanations of the above mathematical formulation in our camera-ready version.
>
> [1] Fan et al, Faceformer: Speech-driven 3d facial animation with transformers, CVPR’22
>
> **Q4**: We suppose the evaluation of control accuracy you mentioned might be two-fold. 1) If it concerns measuring the accuracy of fine-grained local AU control, our proposed CR metric is precise for such evaluation. Since CR is calculated locally for each activated AU on its ARKit-based generated muscle movement, it reflects the effectiveness of fine-grained control over each facial muscle and whether it spills over into adjacent frames. We will add further clarification on Equation 8 to eliminate confusion. Furthermore, our CR metric offers a quantitative basis for future studies to assess whether AUs are precisely activated in fine-grained control tasks. 2) If it concerns whether the locally activated AUs match the semantics of user input, CR cannot evaluate semantic alignment, as this is not its intended purpose. However, because CR does not assess semantic alignment, expression naturalness, or aesthetic quality, we currently evaluate semantic alignment through user studies. In the future, we plan to explore automated evaluation using a neural network.

---

> ### Author Response · Authors · 2024-11-22
> **Response to Reviewer Mxhk (2/2)**
>
> **Q5**: We shall make a clearer illustration of talking styles in Fig. 2 in our camera-ready version. We currently use an "Emo-exempt ID portrait" to represent the talking style, which could lead to misunderstanding. In our manuscript, talking styles are the different facial movement patterns among individuals captured by the CLIP embeddings (green feature vectors in Fig. 2) from facial portraits or caption texts (attributes annotated by hsemotion from the portrait) during training. The talking styles are used to provide a basic facial movement pattern for individuals. We use the newly uploaded video *sup_video_1.mp4* as an example to visualize different talking styles and retargeting between individuals. In that video, the animation with the prompt "older man" tends to express anger by frowning (lowering his eyebrows), while the animation with the prompt "young woman" shows anger by widening her eyes. Additionally, to eliminate further concern, the "A senior actor" in Fig. 1 represents a middle-aged male instead of an occupation.
>
> **Q6**: In our model design, we opted for AUs over ARKit as the fine-grained conditions because it is a conventional practice in previous works (e.g., TalkCLIP [2], InstructAvatar [3]) and industry standards. Moreover, AUs are well-suited for extraction by GPT to train the detector in our approach. For the output format, we chose ARKit due to its compatibility with 3D engines, it is optional and can be replaced by FLAME or other representations. Admittedly, ARKit and AUs share significant semantic overlap, leading to no substantial differences.
>
> [2] Ma et al, TalkCLIP: Talking Head Generation with Text-Guided Expressive Speaking Styles, arXiv:2304.00334
>
> [3] Wang et al, InstructAvatar: Text-Guided Emotion and Motion Control for Avatar Generation, arXiv:2405.15758

---

> > ### Comment · Reviewer_MxhK · 2024-11-23
> > **### Final Rating ###**
> >
> > First I want to thank the other reviewers and authors. My concerns have been addressed and I will marginally increase my score.
> >
> > Soundness: 2 -> 3
> > Presentation: 3
> > Contribution: 3
> > Rating: 6

---

### Official Review · Reviewer_qVk8 · 2024-11-04

**Soundness:** 3
**Presentation:** 2
**Contribution:** 3
**Rating:** 6
**Confidence:** 4

**Summary:**

The paper proposes CafeTalk a diffusion-based method for speech driven 3D talking faces with coarse- and fine-grained multimodal control over facial expressions. CafeTalk has a two-stage training process: in the first stage the model is trained with coarse-grained conditions (e.g. emotions, talking style) using a text encoder. In the second stage it incorporates  fine-grained conditions (localized muscle movements using action units) are learned. A swap-label mechanism is proposed in the second stage to add fine-grained conditional control. Experimental results show that CafeTalk achieves state-of-the-art performance in lip synchronization and expression generation compared to existing methods.

**Strengths:**

1. Novel approach with the use of both coarse- and fine-grained control for 3D talking face generation. This dual-level control allows for more precise and expressive facial animations overcoming the limitations of previous methods that use broader less adaptable controls.
2. Extensive experimentation with evaluations using various datasets and a user study.

**Weaknesses:**

1. The diffusion-based approach for time series combined with the swap-label mechanism in the second stage of training may require substantial computational resources.
2. As noted by the authors the model struggles with lower-face action units when they conflict with speech causing difficulties in achieving accurate synchronization between lip movements and expressions.

**Questions:**

1. A more in-depth discussion of the results is necessary in Sec. 4.2.2 when comparing with the other approaches. For example, why does your model succeed where other models may fall short? It would be interesting to analyze which specific components of your model (e.g., fine-grained control, diffusion model) contribute most to the performance improvements compared to other approaches.

2. What is the inference speed of your approach compared to the others? Can you provide a table comparing inference speeds across methods for a certain input length (e.g. 10 sec) on common hardware along with the number of parameters for each model?

3. W2v2 embeddings contain emotion information. Did you observe any instability during training if the vision and audio emotions contradict?

4. How important is the intensity embedding and the emotion intensity value? Can you include in your ablation study (Sec. 4.2.3) the impact of removing these components on model performance and expression quality?

5. Sec. 3.1.2: You mention that you mask speech audio and coarse-grained conditions with 20% probability. How is the model’s performance affected with this probability? Similarly in Sec. 3.2 you point out “we randomly drop elements triplets (Fd, Fs, Fe) from the AU sequence F” - with what probability?

6. Sec. 4.1: Are the collected YouTube videos ids going to become publicly available? Also, how do you compute the emotion expression and intensity of the collected data?

7. How long can your approach generate frames continuously without degradation in quality? Are there any limitations on sequence length, and can you provide examples for different animation durations (e.g., 3, 6, and 10 seconds)?

8. Table 1: You present results for 3D-ETF benchmark without describing it first in the text. Can you add a description of the 3D-ETF benchmark in the datasets section (Sec. 4.1)?

Minor

Table 2: Correct “Carse” to “Coarse” in the caption.

---

> ### Author Response · Authors · 2024-11-22
> **Response to Reviewer qVk8 (1/3)**
>
> Thank you for your constructive comments and valuable feedback.
>
> **W1**: We intended to emphasize a novel 3D talking face control mechanism with high flexibility and controllability, of which, improving the model’s training efficiency is not our technical focus in this paper. Nevertheless, the training resource of stage 2 is acceptable rather than “substantial” based on our statistics. In stage 2, only the adapter is optimized with just 5.39M parameters. The training time cost is 8% less than that of stage 1. The inference GPU memory occupancy (2480M $\pm$ 42M) and latency (14 $\pm$ 2s) are similar to the ablated model with only stage 1. Despite extra computation, the proposed swap-label mechanism qualifies as a contribution to achieving fine-grained details control, and the training cost is acceptable on the current computational platform. Moreover, it is worth noting that diffusion-based architecture is proven to achieve superior performance (ref. Audio2Photoreal [1], Media2Face [2]), although the diffusion model has its known challenge of training efficiency, it is not the concentration of our proposed approach.
>
> [1] Ng et al, From Audio to Photoreal Embodiment: Synthesizing Humans in Conversations, CVPR’24
>
> [2] Zhao et al, Media2Face: Co-speech Facial Animation Generation with Multi-Modality Guidance, SIGGRAPH’24
>
> **W2**: We understand that the reviewer considers the technical limitation mentioned in our paper a weakness. However, we mentioned the decline of metric values under explicit conflict user inputs as a limitation only from a technical point of view rather than a “struggling” (as mentioned in the question) subpar model performance. As a user-centric task of controlling the 3D talking face, we believe a well-designed model should align with the user’s intent and elevate the priority of user input. Although the expression and speech could be different due to the user's interference, our model can still incorporate conflict inputs and, in most cases, ensure the output matches their expectations, where the user probably has valid reasons. Furthermore, our model’s performance of lip synchronization with fine-grained manipulation is widely accepted by the participants in our user study, which validates our contribution to enhancing flexible and controllable user-centric generation.
>
> **Q1**: The advantages of our model in the evaluation metrics mainly stem from two factors: (1) the structure of the diffusion model, which significantly enhances the diversity of output (as reflected in the Div metric) compared to discriminator-based models, while avoiding the over-smoothing issue (LVE); and (2) the use of AdaLN for coarse-grained condition injection, which has proven [3] to be more effective than previous methods in injecting and maintaining semantic control (as shown in the Acc metric, Tab. 2).
>
> [3] Peebles et al, Scalable diffusion models with transformers, ICCV’23
>
>
>
>
>
>
>
>
>
>
> **Q2**: As shown in the table below, we report the parameter count of the relevant models and the inference time for 5-second audio on a 2080Ti. We shall add this table in the camera-ready version of the appendix in the spirit of the page limitation.
> | Models      |    Num of parameters (M)    |    Inference time (sec)    |
> |-------------|:---------------------------:|:--------------------------:|
> | FaceFormer  |            94.7             |           3.34             |
> | TalkSHOW    |            94.9             |           3.69             |
> | EMOTE       |           109.6             |           3.73             |
> | EmoTalk     |           640.6             |          10.99             |
> | UniTalker   |            96.0             |           3.20             |
> | Cafe-Talk   |           345.9             |          14.71             |
>
> The inference latency is primarily caused by the sampling and denoising process of the current diffusion architecture, which could be partially alleviated with a more advanced GPU. In real-world applications, our method's high controllability, precise manipulation, and diverse outputs are compatible with flexible offline facial editing and time-tolerant AIGC talking animation generation.

---

> ### Author Response · Authors · 2024-11-22
> **Response to Reviewer qVk8 (2/3)**
>
> **Q3**: We did not encounter instability in either training stage. In stage 1, the audio and emotion labels from the MEAD or RAVDESS datasets are paired, ensuring no conflicting emotions during training. The emotion label serves as an explicit emotion condition, which the model prioritizes and learns as the dominant factor (see table below). In stage 2, although conflicts may exist between audio and vision emotions (emotion labels), the emotion label takes precedence over audio. It is qualitatively proven by an extra test in the following table, indicating that the audio provides less influential information for emotion semantics without interfering with the training stability. Additionally, since only the adapter’s local control ability is trained in stage 2, the unpaired data does not affect the original capability of the base model.
> |  |  Acc   |  Div  |
> |------|:-----:|:-----:|
> | Mask emotion label            |           23.68%            |          110.96           |
> | Swap emotion label            | 57.22% (label as gt), 10.86% (audio as gt)     |          106.33           |
> | Paired emotion label (Tab 1)  |           59.48%            |          119.36           |
>
> **Q4**: We shall add more ablation analysis on the intensity control. We provide the following details here for better understanding. On the MEAD test set, higher intensity levels result in higher Acc, indicating that incorporating expression intensity enhances the emotional expressiveness of the generated results. When intensity is set to 0, we observe a sharp drop in both Acc and Div, demonstrating that emotion label control fails in this extreme case and confirming that our design enables the model to learn the ordinal relationship of intensity. In an ablation study removing intensity from the pipeline, we observed a decline in both accuracy and diversity, suggesting the model is more likely to generate expressions with medium intensity. This conclusion further highlights the necessity of intensity for flexible control.
>
> | |Acc  |  Div  |
> |-----------|:--------:|:---------:|
> | MEAD testset int = 1 subset    |           51.68             |          111.16           |
> | MEAD testset int = 2 subset    |           63.86             |          119.75           |
> | MEAD testset int = 3 subset    |           66.26             |          119.48           |
> | intensity = 0                  |           19.07             |           67.09           |
> | remove intensity               |           53.27             |          115.17           |
> | MEAD testset (Tab 1)               |           59.48%            |          119.36           |
>
> **Q5**: Random masking during training enhances the expressiveness of the generated movements during inference. Empirically, we set the random dropping probability to 20%. A lower probability makes the CFG technique ineffective, while a too-high probability may lead to unsynchronized lip movements. This probability is similar to those used in Audio2Photoreal [1] and MDM [4]; We expect users to control facial movements globally using coarse-grained conditions, with only few fine-grained controls for adjustment. Therefore, fine-grained conditions should remain sparse during inference. To align with this, we set the random dropping probability for triplets to 80%.
>
> [4] Tevet et al, Human Motion Diffusion Model, ICLR’23

---

> ### Author Response · Authors · 2024-11-22
> **Response to Reviewer qVk8 (3/3)**
>
> **Q6**: We plan to release the collected 157-hour dataset after the paper is accepted. This dataset was used exclusively in stage 2 training to enhance the model's generalization ability. Since the swap-label mechanism generates conflicting coarse- and fine-grained conditions, no labeling was required as the emotion labels and intensity values were randomly assigned.
>
> **Q7**: Our model is trained on data with durations ranging from 3 to 10 sec. Longer inputs are first segmented, and each segment is generated separately and then concatenated with interpolation. This approach ensures stable results and faster inference. Demo videos of varying durations (4 to 10 sec) are provided in the supplementary materials. In future work, we plan to incorporate historical movements as conditions to guide the model's generation, inspired by related co-speech work such as GestureDiffuCLIP [5].
>
> [5] Ao et al, GestureDiffuCLIP: Gesture Diffusion Model with CLIP Latents, SIGGRAPH’23
>
> **Q8**: The 3D-ETF benchmark is the test set of RAVDESS provided by Peng et al mentioned in Ln 356. We apologize for the inconvenience and will ensure consistency in our camera-ready version.

---

> ### Comment · Reviewer_qVk8 · 2024-11-25
> **Review Update: Response to Rebuttal**
>
> I would like to thank the authors for addressing my concerns. I am updating my rating from 5 to 6.

---

### Meta-Review · Area_Chair_v68j · 2024-12-21

**Metareview:**

This paper introduces a novel approach for 3D talking face generation that incorporates both coarse- and fine-grained controls. This method allows for more precise and expressive facial animations, surpassing the limitations of previous methods that relied on less adaptable controls. The authors validate their framework through extensive experiments on multiple datasets and a user study. Reviewers initially raised concerns regarding the experimental results, including the need for a more in-depth discussion, an evaluation of inference speed, and the presence of abrupt expression shifts. The authors properly addressed these concerns in their rebuttal.  The paper received an overall positive rating from reviewers, and I concur with their review, recommending acceptance.

The authors should incorporate the clarifications and new experiments into the revised paper.

**Additional Comments On Reviewer Discussion:**

Three reviewers actively engaged with the authors' responses during the rebuttal process.  Initial concerns centered on the completeness of the experiments. The authors addressed these concerns by providing new experiments on inference speed, clarifying real-world use cases for dramatic emotion changes, and presenting additional examples with multiple facial muscles. After rebuttal, all three reviewers maintained or increased their scores, resulting in an overall positive assessment of the paper's strengths.

---

### Decision · Program_Chairs · 2025-01-22

Accept (Poster)